# Evaluation of the Abbott BinaxNOW COVID-19 Test Ag Card for rapid detection of SARS-CoV-2 infection by a local public health district with a rural population

**Rachel E. Pollreis**[1]*, **Clay Roscoe**[1☯], **Rachel J. Phinney**[1☯], **Surabhi S. Malesha**[1☯], **Matthew C. Burns**[2‡], **Aimee Ceniseros**[2‡], **Charles H. Washington**[1], **Andrew J. Nutting**[1], **Christopher L. Ball**[2]

1 Southwest District Health, Caldwell, Idaho, United States of America, 2 Division of Public Health, Idaho Department of Health and Welfare, Boise, Idaho, United States of America

☯ These authors contributed equally to this work.
‡ These authors also contributed equally to this work.
* Rachel.Pollreis@phd3.idaho.gov

**Data Availability Statement:** De-identified patient level data cannot be shared publicly because of legal restrictions. These data were collected by

## Abstract

SARS-CoV-2 RT-PCR, the gold standard for diagnostic testing, may not be readily available or logistically applicable for routine COVID-19 testing in many rural communities in the United States. In this validation study, we compared the BinaxNOW™ COVID-19 Test Ag Card with SARS-CoV-2 RT-PCR in 214 participants who sought COVID-19 testing from a local public health district in Idaho, USA. The median age of participants was 35 and 82.7% were symptomatic. Thirty-seven participants (17.3%) had positive RT-PCR results. Results between the two tests were 94.4% concordant. The sensitivity of the BinaxNOW™ COVID-19 Test Ag Card was 67.6% (95% CI: 50.2–81.9%), and the specificity was 100.0% (95% CI: 97.9–100.0%). The positive predictive value (PPV) for the BinaxNOW™ COVID-19 Test Ag Card was 100.0% (95% CI: 86.2–100.0%), and the negative predictive value (NPV) was 93.6% (95% CI: 89.1–96.6%). Although the sensitivity of BinaxNOW™ COVID-19 Test Ag Card was lower than RT-PCR, rapid results and high specificity support its use for early detection of COVID-19, especially in settings where SARS-CoV-2 RT-PCR testing is not readily available. Rapid antigen tests, such as the BinaxNOW™ COVID-19 Test Ag Card, may be a more convenient tool in quickly identifying and preventing COVID-19 transmission, especially in rural settings.

## Introduction

Early detection of SARS-CoV-2, the virus that causes COVID-19, is essential for slowing community transmission. Real-time reverse transcription-polymerase chain reaction (RT-PCR) assays for COVID-19 diagnosis have excellent sensitivity and specificity but require laboratory instrumentation and sending specimens to a laboratory for testing can culminate in delayed turnaround times for results [1]. Rapid, point-of-care tests that quickly identify individuals

Southwest District Health as part of reportable disease surveillance under Idaho administrative code (IDAPA 16.02.10 https://adminrules.idaho.gov/rules/current/16/160210.pdf). Use of these data for other purposes requires approval from the Idaho Division of Public Health. De-identified patient data can be requested from the Idaho Division of Public Health by contacting the Bureau of Communicable Diseases Epidemiology Section at Epimail@dhw.Idaho.gov. Other types of data included in this paper are publicly available. SARS-CoV-2 genomic sequencing is available through the global initiative on sharing influenza data (GISAID.org). Specific sequence accession IDs used in this paper can be provided on request by contacting the Bureau of Laboratories at IBLsequencing@dhw.idaho.gov.

**Funding:** The authors received no specific funding for this work.

**Competing interests:** The authors have declared that no competing interests exist.

with SARS-CoV-2 are particularly useful in public health settings to limit the spread of infection [2]. The BinaxNOW™ COVID-19 Test Ag Card (BinaxNOW™ COVID-19 Test) is a rapid antigen test in a card format that detects SARS-CoV-2 protein antigens present in a nasal swab specimen. This test received an Emergency Use Authorization (EUA) from the Food & Drug Administration (FDA) on August 26, 2020 [3]. The BinaxNOW™ COVID-19 Test is a lateral flow immunoassay used at the point of care with results read visually after a 15-minute incubation period. Use of the BinaxNOW™ COVID-19 Test has increased the availability of COVID-19 testing in Idaho, where laboratory-based COVID-19 testing remains limited for many geographically isolated, rural communities.

Additional data on the performance of the BinaxNOW™ COVID-19 Test are needed to help providers interpret results and determine whether confirmatory testing should be recommended [4]. A prospective, observational study at a local public health district in Southwestern Idaho was conducted to evaluate the performance of BinaxNOW™ COVID-19 Test compared with the gold standard of SARS-CoV-2 RT-PCR. Demand for COVID-19 testing was evaluated over time, by monitoring testing rates throughout the region as well as within the Southwest District Health clinic.

## Materials and methods

Through this evaluation, we determined the percent concordance, sensitivity, specificity, positive predictive value, and negative predictive value of BinaxNOW™ COVID-19 Test and SARS-CoV-2 RT-PCR. Examination of BinaxNOW™ COVID-19 Test performance characteristics employed a modified, more restrictive positive test interpretation criteria, where only bands that extended across the full width of the test strip were scored as positive, as recommended by Pilarowski and colleagues [5].

### Study site and participant enrollment

This study was conducted by Southwest District Health, one of Idaho's seven public health districts, that serves six counties with a total estimated population of 283,930 residents. All counties in Southwest District Health are at least partially classified as rural by the Health Resources & Services Administration's Federal Office of Rural Health Policy [6]. Subjects were identified among clients who scheduled an appointment for COVID-19 testing with Southwest District Health Employee Health/Clinic Services in Caldwell, Idaho, or through any mobile testing locations facilitated by Southwest District Health between November 2020 and May 2021. Prior to enrollment and performance of any study-specific procedure, a signed informed consent form was obtained for each participant. Interpreter services were offered when a language barrier was present. For minors, written consent from a parent or guardian was required for enrollment, as well as verbal assent from the minor. Eligible participants were those who sought COVID-19 testing from the Southwest District Health Clinic, were able to provide informed consent, and had not been diagnosed with COVID-19 in the 90 days prior to specimen collection. All participants were screened using a standardized questionnaire to collect demographic data and to identify symptom, exposure, and immunization status. Demographic data including age, gender, race, ethnicity, and contact information were collected and participants were asked about their symptoms and previous vaccinations, recent travel, and whether they had been exposed to a person diagnosed with COVID-19 in the previous 14 days.

### Specimen collection and testing procedures

COVID-19 testing using the BinaxNOW™ COVID-19 Test was conducted according to the manufacturer's instructions. A nasal swab was collected by inserting the swab into the nostril

until resistance is met at the level of the turbinate, rotating five times against the nasal wall, and repeating in the other nostril using the same swab [7]. Results were visually read promptly 15 minutes after the test card was closed. A second specimen was collected from each participant for SARS-CoV-2 RT-PCR testing, using the same swab collection procedure. Second specimens were immediately placed in 3 mL virus stabilization tubes containing viral transport media and stored between 2°C −8°C until transported to the Idaho Bureau of Laboratories (IBL) for RT-PCR testing using the TaqPath™ COVID-19 Combo Kit (ThermoFisher Scientific A47814). Results were determined by the automated ThermoFisher Interpretive Software™ and confirmed by IBL staff. Results were available in 1–3 days of specimen collection. A subset of positive specimens was sequenced with the Oxford Nanopore Technologies (ONT) MinION Mk1c using the ONT protocol "PCR tiling of SARS-CoV-2 virus with rapid barcoding (SQK-RBK110.96) Version: PCTR_9125_v110_revD_24Mar2021" developed by Freed and Silander [8]. Assemblies were performed using CLC Genomics version 21.0.4 long read support module and lineage calls made by the Pangolin COVID-19 Lineage Assigner version 3.1.5 [9]. The resulting consensus sequences were uploaded to GISAID [10].

## Data governance and analysis

A unique study identification number was assigned to each participant in the order of enrollment. Paper screening forms were archived and stored in a locked filing cabinet, while results from RT-PCR testing were reported using an electronic portal and added to the study database. The study database is stored securely on Southwest District Health servers, with access restricted to study personnel. Data analysis was conducted using Excel for data storage, with R 4.0.3 software used for statistical analysis.

Sample size calculations were based on the primary outcome, percent concordance of BinaxNOW™ COVID-19 Test and SARS-CoV-2 RT-PCR results, using precision-based calculations. For an estimated concordance of 90%, a sample size of 200 participants would produce a two-sided 95% confidence interval of width ± 4.2%. Secondary outcomes include determining the sensitivity, specificity, positive predictive value, and negative predictive value for the BinaxNOW™ COVID-19 Test, which are reported as percentages with two sided 95% confidence intervals using an alpha of 0.05.

COVID-19 testing demand was monitored throughout the study enrollment window by the Idaho Department of Health & Welfare. Data are based on electronic laboratory reports (ELRs) received from laboratories that report both positive and negative results. Data are also based on the date the specimen was collected, not the date the lab result was received. Records were extracted weekly and shared with Idaho's Public Health Departments.

## Ethics and confidentiality

This project was reviewed by the Idaho Department of Health and Welfare, Division of Public Health Research Determination Committee and was deemed public health surveillance activity. The project was determined as non-research and exempt from ethical review by the Institutional Review Board.

Participant confidentiality is strictly held in trust by the participating investigators, staff, and institutions. All research staff received training in Health Insurance Portability and Accountability Act (HIPAA) and privacy and confidentiality as part of their routine duties. Participant information was stored securely, de-identified, and used for analysis as described above. Documents that identify the participant (e.g., the signed informed consent) were maintained in a locked file cabinet at Southwest District Health with access limited to the principal investigator.

## Results

Between November 2020 and May 2021, 214 eligible participants were enrolled in the study and paired nasopharyngeal samples were collected and tested. The majority (66.4%, n = 142) were female. Participant ages ranged from 5 to 95 years of age, with a median (IQR) of 35 (19–55) years; 22.4% (n = 48) of participants were under the age of 18. Samples were collected from participants of varying races and ethnicities, most of whom were white (65.9%, n = 141). Nearly one-third of participants (28.9%, n = 62) did not disclose race or ethnicity. Additional demographics data can be found in the S1 Table.

Of the 214 participants, 17.3% (n = 37) had positive SARS-CoV-2 RT-PCR detection and 11.6% (n = 25) had positive results with the BinaxNOW™ COVID-19 Test. The concordance between the BinaxNOW™ COVID-19 Test and SARS-CoV-2 RT-PCR, was 94.4% (n = 202). The BinaxNOW™ COVID-19 Test demonstrated a sensitivity of 67.6% (95% CI: 50.2–81.9%), and specificity of 100.0% (95% CI: 97.9–100.0%). Compared to RT-PCR, the positive predictive value (PPV) for the BinaxNOW™ COVID-19 Test was 100.0% (95% CI: 86.2–100.0%), and the negative predictive value (NPV) was determined to be 93.6% (95% CI: 89.1–96.6%).

Of the 214 participants, 82.7% (n = 177) reported symptoms at the time of sample collection. The median (IQR) days since symptom onset was 2 (1–3) days prior to sample collection. The most frequently reported symptoms were congestion/runny nose (n = 123), headache (n = 121), fatigue (n = 116), sore throat (n = 116), and cough (n = 116). Only 25.2% (n = 54) of participants reported a fever as one of their symptoms. The prevalence of COVID-19 among all study participants was 17.3%, which was lower than among symptomatic participants (22.9%) but higher than among asymptomatic participants (12.1%).

Of the 214 paired samples, 5.6% (n = 12) had discordant results. All discordant results were due to a false negative from the BinaxNOW™ COVID-19 Test. A total of 83.3% (n = 10) of participants with discordant results reported symptoms at the time of sample collection. The median (IQR) days between symptom onset and sample collection for symptomatic participants with discordant results were 2.5 (1–4) days (n = 10), with a greater variability in days since symptom onset compared with symptomatic participants with concordant results with a median (IQR) of 2 (1–3) days (n = 167) since symptom onset (Fig 1). There was no significant difference in days since symptom onset when comparing concordant and discordant results (p = 0.439) using the student's t-test (Table 1).

A minority of participants were asymptomatic at the time of sample collection (17.3%, n = 37). Of the asymptomatic participants, 37.8% (n = 14) reported a known exposure to someone who had tested positive for COVID-19 within the last 14 days and 10.8% (n = 4) were determined by RT-PCR to be positive for COVID-19. The BinaxNOW™ COVID-19 Test demonstrated a sensitivity of 50.0% (95% CI: 6.8–93.2%), and specificity of 100.0% (95% CI: 89.4–100.0%) for asymptomatic participants.

A total of 14.5% of participants received at least one dose of a COVID-19 vaccination, and 12.1% of participants were fully immunized. Of those who were fully immunized, 70.3% (n = 19) showed no symptoms. One fully immunized participant (3.7%, n = 1), who did not report any symptoms at time of sample collection, tested positive for COVID-19 through RT-PCR with concordant BinaxNOW™ COVID-19 Test results.

The Idaho Bureau of Laboratories sequenced 33 of the 37 positive samples (Table 2). Of the 33 samples successfully sequenced, 25 produced unambiguous lineage calls and 8 produced sequences with some ambiguity as assigned by PangoLEARN [9]. During the enrollment period, the most prevalent SARS-CoV-2 variants were Alpha (B.1.1.7), Beta (B.1.351), and Epsilon (B.1.427 & B.1.429).

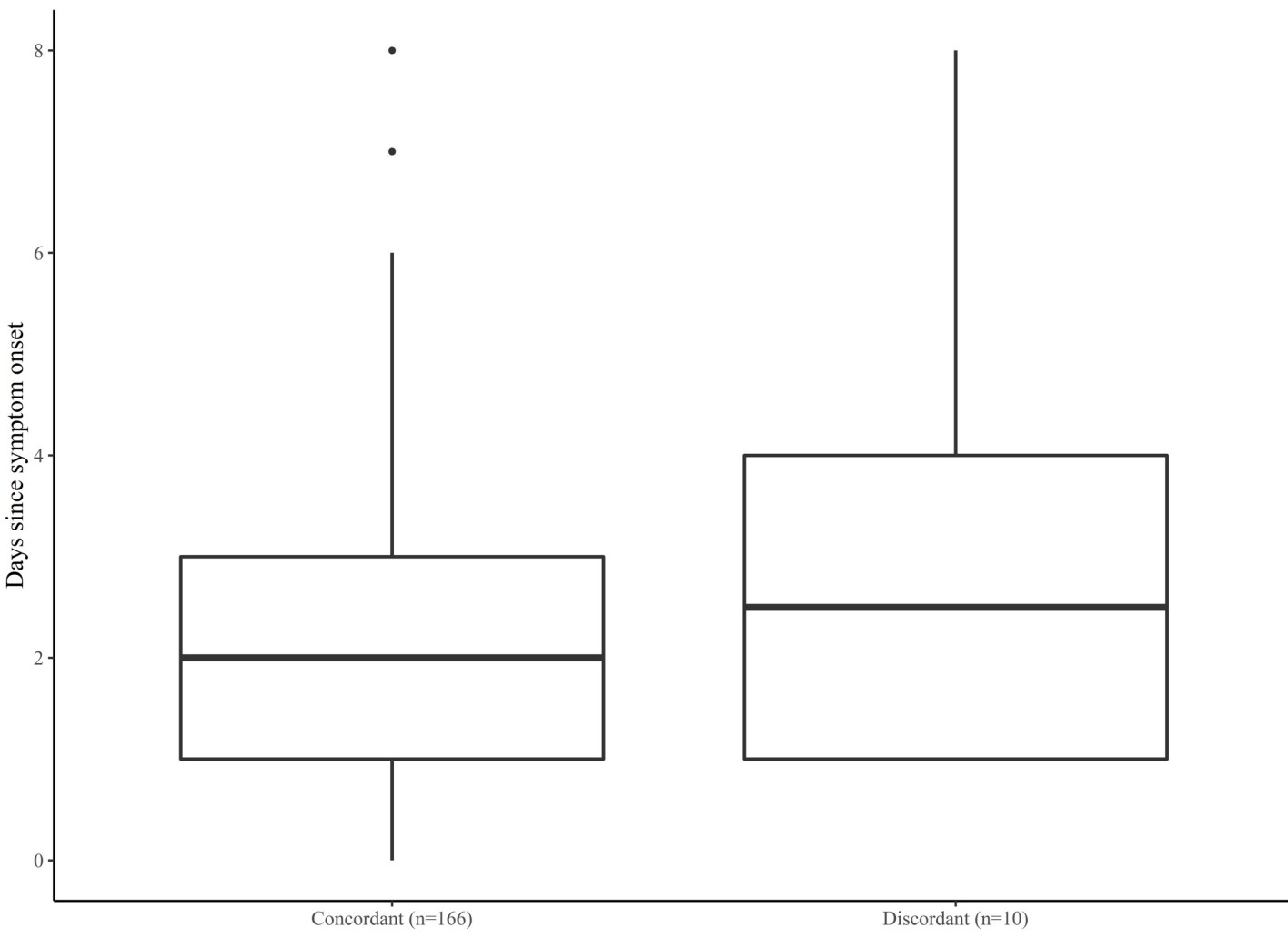

**Fig 1. Days between symptom onset and sample collections for participants with concordant and discordant results.**

Of the 37 RT-PCR positive specimens collected, the cycle threshold (Ct) values ranged from 16 to 38, with a median value of 28 (Table 3). The median Ct value for samples with concordant results was 24, compared to a median Ct value of 36 for discordant samples (p<0.001) (Table 4). For samples with a Ct value less than 35, the BinaxNOW™ COVID-19 Test Ag Card demonstrated a sensitivity of 85.7% (95% CI: 67.3–95.9%).

Throughout the enrollment period, Southwest District Health clinic saw a median (IQR) RT-PCR turnaround time of 1 (0–2) days. During the same time period, the region as a whole saw a median wait time of approximately 2 (1–4) days for RT-PCR results to reach the patient. The discrepancy in turnaround time is likely due to the Southwest Health District Clinic

**Table 1. Comparison of days since symptom onset for concordant and discordant results using the student's t-test.**

| | | Days Since Symptom Onset | | | | |
|---|---|---|---|---|---|---|
| Results | n | Median | IQR | t | df | p-value |
| Concordant | 167 | 2 | (1–3) | | | |
| Discordant | 10 | 2.5 | (1–4) | -0.775 | 174 | 0.439 |

**Table 2. SARS-CoV-2 lineage calls shown by concordance between BinaxNOW™ COVID-19 Test and RT-PCR.**

| Lineage | Discordant | Concordant | Total |
|---|---|---|---|
| B.1 | 1 | 1 | 2 |
| B.1.1.7 *(Alpha, VOC[1])* | 3 | 0 | 3 |
| B.1.1.416 | 0 | 1 | 1 |
| B.1.2 | 5 | 13 | 18 |
| B.1.170 | 1 | 0 | 1 |
| B.1.234 | 0 | 1 | 1 |
| B.1.400 | 0 | 1 | 1 |
| B.1.427 *(Epsilon, VOI[2])* | 0 | 1 | 1 |
| B.1.429 *(Epsilon, VOI)* | 0 | 2 | 2 |
| B.1.544 | 1 | 0 | 1 |
| B.1.561 | 0 | 1 | 1 |
| B.1.596 | 0 | 1 | 1 |
| **Total Variants Identified** | 11 | 22 | 33 |

[1]Variant of Concern (VOC) is classified by the CDC as a variant for which there is evidence of increased transmissibility, more severe disease, significant reduction in neutralization by antibodies generated during previous infection or vaccination, reduced effectiveness of treatments or vaccines, or diagnostic detection failures [11].
[2]Variant of Interest (VOI) is classified by the CDC as a variant with specific genetic markers that have been associated with changes to receptor binding, reduced neutralization by antibodies generated against previous infection or vaccination, reduced efficacy of treatments, potential diagnostic impact, or predicted increase in transmissibility or disease severity [11].

having immediate access to Electronic Lab Reports (ELRs) while many private clinics throughout the region relied on private web based patient portals or mailed RT-PCR results to inform patients of their results.

Participant enrollment varied during the study period (Fig 2). Study enrollment paused between February 6, 2021 and March 25, 2021 (MMWR week 6–12) due to leadership staff turnover, new training and implementation, as well as prioritization of vaccination efforts. During the study period, the demand for COVID-19 testing across the health district declined over time.

## Discussion

The BinaxNOW™ COVID-19 Rapid Antigen Test was evaluated in people seeking free testing conducted by a local public health district in Idaho serving a rural population. Results were compared to the gold standard of SARS-CoV-2 detection by RT-PCR. The prevalence of COVID-19 among all study participants was 17.3%, indicative of high levels of community transmission of SARS-CoV-2 during the study time frame. Overall, results were highly concordant (94.4%) and no false positives were identified. Sensitivity of the BinaxNOW™ COVID-19 Test was 67.6% overall, and lower for asymptomatic participants. Results from our study were

**Table 3. Cycle threshold (Ct) values shown by concordance between BinaxNOW™ COVID-19 Test and RT-PCR.**

| Ct value | Concordant | Discordant | Total |
|---|---|---|---|
| *<30* | 20 | 2 | 22 |
| *31–35* | 4 | 3 | 7 |
| *>35* | 1 | 7 | 8 |

**Table 4. Comparison of cycle threshold (Ct) values for concordant and discordant results using the Welch's t-test for unequal variance.**

| Results | n | Median | IQR | t | df | p-value |
|---|---|---|---|---|---|---|
| | | | Cycle threshold (Ct) values | | | |
| Concordant | 25 | 24.0 | (20–29.5) | | | |
| Discordant | 12 | 36.0 | (32.75–38) | -5.38 | 27.96 | <**0.001** |

consistent with larger studies assessing the performance of the BinaxNOW™ COVID-19 Test. A study conducted by in Pima County, Arizona in November of 2020 found similar results, with a 64.2% sensitivity for symptomatic participants, 35.8% sensitivity for asymptomatic participants, and a 100% specificity for all participants regardless of symptom status [12]. Another study evaluating drive-through COVID-19 testing in Massachusetts demonstrated the Binax-NOW™ COVID-19 Test to have a similar specificity of 100% for symptomatic adults and 99.6% for asymptomatic adults. However, a higher sensitivity for symptomatic adults (96.5%), compared to asymptomatic adults (70.2%) was observed. The Massachusetts study also demonstrated that sensitivity was significantly lower when BinaxNOW™ COVID-19 Tests were

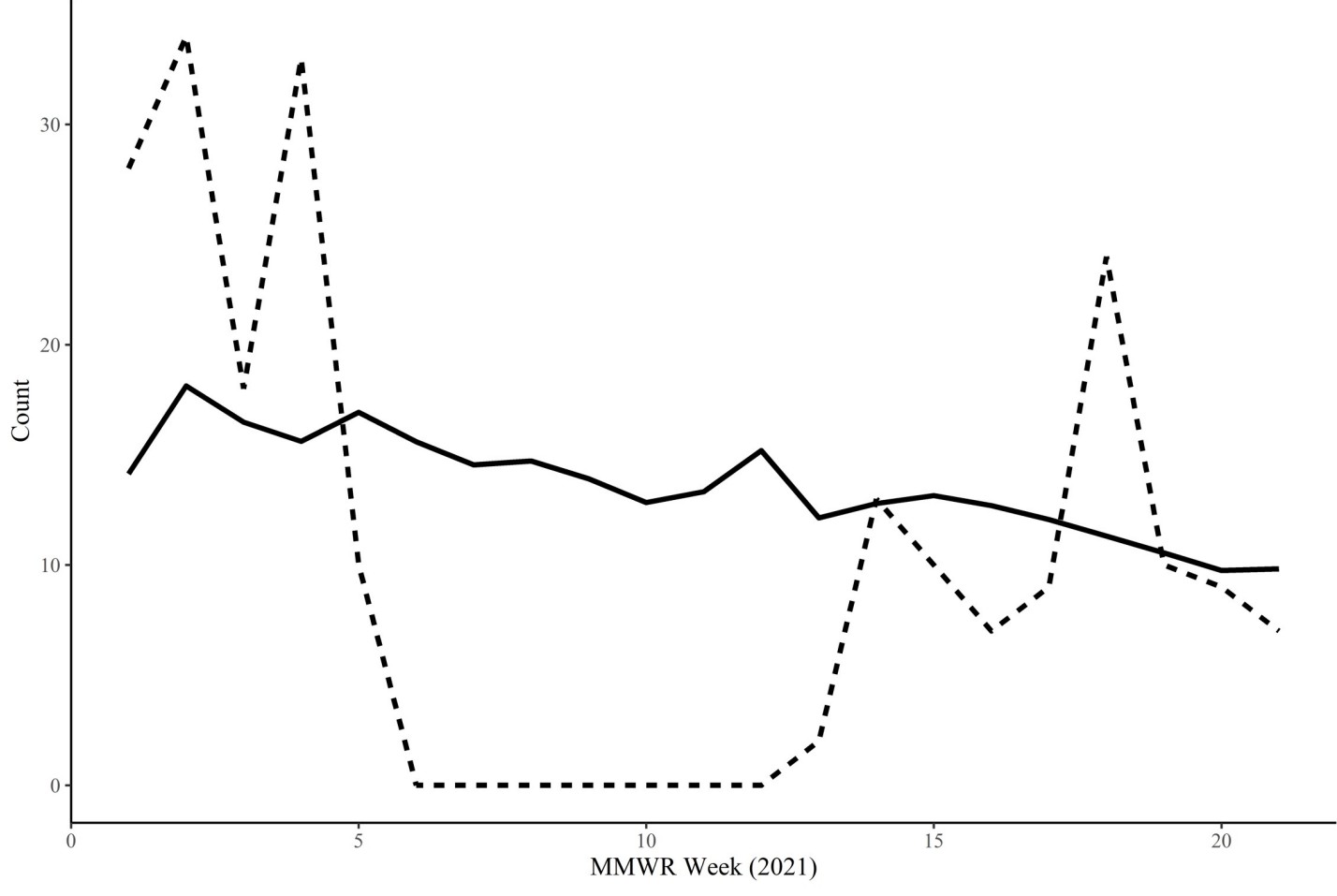

**Fig 2. COVID-19 testing demand in SWDH (per 1,000 residents) and newly enrolled study participants by week, Jan-May 2021.**

conducted at temperatures below the manufacturer's recommended range [13]. Although BinaxNOW™ COVID-19 Tests were conducted indoors for our study, samples were collected in an outdoor drive through tent; the indoor testing area was close to a doorway leading to the outdoor testing site, and temperatures, from sample collection to test results, were not monitored.

One objective of our study was to inform recommendations on whether confirmatory testing by RT-PCR is needed following rapid antigen testing with the BinaxNOW™ COVID-19 Test. The lack of false positives observed in this study, along with specificity above 99% reported from other studies, indicate that positive results by BinaxNOW™ COVID-19 Test may not require confirmation by RT-PCR. However, lower sensitivity and the possibility of a false negative result strongly supports that confirmatory testing should be recommended in some circumstances, especially when the results of the BinaxNOW™ COVID-19 Test is inconsistent with the clinical context (e.g. among symptomatic individuals or those with a known exposure to someone with COVID-19), as per current CDC recommendations [14].

The main benefits of rapid antigen testing are its accessibility, and that it can be conducted outside of a laboratory setting with rapid availability of results at the point of care. In our study, the median turnaround time for RT-PCR results was approximately 24 hours, though early in the pandemic, rural areas of Idaho experienced significant turnaround times for RT-PCR results, with most results taking more than a week [15]. The BinaxNOW™ COVID-19 Test might be particularly useful in rural settings without local RT-PCR capacity.

Among symptomatic participants with false negatives from the BinaxNOW™ COVID-19 Test, we saw a greater variability in days since symptom onset (see Fig 1), however differences in days since symptom onset did not differ between discordant and concordant samples, likely due to a small number of symptomatic participants with discordant results (n = 10). Because viral loads are generally higher at symptom onset and decline over time, the BinaxNOW™ COVID-19 Test might be less able to detect SARS-CoV-2 in individuals with specimen collection several days after symptom onset. Additionally, the BinaxNOW™ COVID-19 Test was less sensitive among asymptomatic participants, whereas viral loads have been shown to be comparable between symptomatic and asymptomatic COVID-19 patients [16]. In this study, 62.1% of asymptomatic participants did not report a known exposure in the 14 days prior to sample collection. Since asymptomatic participants without a known exposure to COVID-19 are unable to define a timeline of their exposure, we can assume there is greater variability in the days since exposure, and consequently, will have a greater variability in SARS-CoV-2 viral load [17]. Generally, rapid antigen tests perform better when viral loads are high, which is also when people with SARS-CoV-2 are more likely to be infectious.

Demand for COVID-19 testing declined rapidly during our enrollment period. There was a decline in SARS-CoV-2 RT-PCR testing demand throughout Idaho as well as a decline in new participants. We can attribute this to a decline in COVID-19 prevalence throughout the region and an increase in COVID-19 immunization during this time.

There were several limitations to this study, including small sample size, which may have affected analysis of the performance of the BinaxNOW™ COVID-19 Test, compared to RT-PCR. Participants were identified among people seeking testing from the local public health district and they may not be representative of the community as a whole. An additional limitation may have been the temperature variability of the testing location, which may have affected performance of the BinaxNOW™ COVID-19 Test, though the primary methodology of COVID-19 testing (rapid antigen and RT-PCR) in the United States mirrors this process, with samples being collected from patients outdoors or within their personal vehicle and this study may inform these practices.

Rapid antigen testing, including the BinaxNOW™ COVID-19 Test, is an important tool in detecting SARS-CoV-2, particularly in rural areas with limited access to high level laboratories that are capable of conducting SARS-CoV-2 RT-PCR. While the BinaxNOW™ COVID-19 Test is less sensitive than SARS-CoV-2 RT-PCR, it is highly specific, and can be used as a tool to rapidly detect SARS-CoV-2 in community settings. Community testing programs run by local health departments using rapid antigen tests can improve access to COVID-19 testing and to help limit the spread of SARS-CoV-2. As additional waves of novel coronavirus variants emerge, rapid antigen testing may continue to serve as a valuable alternative testing modality to RT-PCR tests.

## Supporting information

**S1 Table. Characteristics of study participants and SARS-CoV-2 detection method and result.**
(DOCX)

## Acknowledgments

The authors want to express their utmost gratitude to Dr. Eileen Dunne for her contributions to this research and article. The authors also would like to express their gratitude to the members of the Idaho National Guard, who were assigned to Southwest District Health, for assisting in sample collection as well as Dr. Nikole Zogg, Director of Southwest District Health for her support of this research. We would also like to thank the residents of Southwest District Health and those who participated in this research.

## Author Contributions

**Conceptualization:** Clay Roscoe, Christopher L. Ball.

**Data curation:** Rachel E. Pollreis, Rachel J. Phinney, Matthew C. Burns, Andrew J. Nutting.

**Formal analysis:** Rachel E. Pollreis, Matthew C. Burns, Aimee Ceniseros.

**Investigation:** Clay Roscoe, Rachel J. Phinney, Surabhi S. Malesha, Andrew J. Nutting.

**Resources:** Aimee Ceniseros.

**Supervision:** Clay Roscoe, Rachel J. Phinney, Surabhi S. Malesha, Andrew J. Nutting, Christopher L. Ball.

**Visualization:** Rachel E. Pollreis.

**Writing – original draft:** Rachel E. Pollreis.

**Writing – review & editing:** Rachel E. Pollreis, Clay Roscoe, Rachel J. Phinney, Charles H. Washington.

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
