## [Decision Letter · Decision Letter 0]

17 Sep 2021

PONE-D-21-27586Evaluation of the Abbott BinaxNOW™ COVID-19 Test Ag Card for rapid detection of SARS-CoV-2 infection by a local public health district with a rural populationPLOS ONE

Dear Dr. Pollreis,

Thank you for submitting your manuscript to PLOS ONE. After careful consideration, we feel that it has merit but does not fully meet PLOS ONE’s publication criteria as it currently stands. Therefore, we invite you to submit a revised version of the manuscript that addresses the points raised during the review process.

We look forward to receiving your revised manuscript.

Kind regards,

Shinya Tsuzuki, MD, MSc

Academic Editor

PLOS ONE

Journal Requirements:

Additional Editor Comments:

All reviewers evaluated that the manuscript was appropriate for publication and I agree with their opinion. A few comments should be addressed still remained then please respond them.

Reviewers' comments:

Reviewer's Responses to Questions

**Comments to the Author**

1. Is the manuscript technically sound, and do the data support the conclusions?

Reviewer #1: Yes

Reviewer #2: Yes

Reviewer #3: Yes

2. Has the statistical analysis been performed appropriately and rigorously? 

Reviewer #1: Yes

Reviewer #2: I Don't Know

Reviewer #3: Yes

3. Have the authors made all data underlying the findings in their manuscript fully available?

Reviewer #1: Yes

Reviewer #2: Yes

Reviewer #3: Yes

4. Is the manuscript presented in an intelligible fashion and written in standard English?

Reviewer #1: Yes

Reviewer #2: Yes

Reviewer #3: Yes

5. Review Comments to the Author

Reviewer #1: I have no comments on the submitted manuscript "Evaluation of the Abbott BinaxNOW™ COVID-19 Test Ag Card for rapid detection of SARS-CoV-2 infection by a local public health district with a rural population"

Reviewer #2: The study describes the performance of the BinaxNOW Ag Card for the detection of SARS-CoV-2 virus antigen in clinical samples. The manuscript is well-structured and readily understandable, and the authors conclusions that are drawn from the presented data are plausible. The results are discussed from several perspectives and integrated into current knowledge about the BinaxNow test performance. However, discussing the BinaxNOW test performance with that of other rapid test products would add further value to the manuscript.

In ll270ff the authors state that, due to the high specifity of the BinaxNOW test, positive test results may require confirmation by RT-PCR. I do not understand why, please explain. A positive result of a high specificity test should be ok?

Other points:

Due to the diverse test princriples in point of care testing, the precise format (lateral flow assay?) would be helpful

Beginning with citation #10, the numbering is flawed. Please correct

Tables 2+3: <total identified="" variants=""> is somewhat misleading, you could consider to choose another term (e.g. <total identified="" viruses=""> or just <total>)

Is the information in Table 3 relevant for the interpretation or discussion of study results?

Tables 4+5: in the text 25 concordant samples are described, in Tables 4+5 only 24 samples are listed</total></total></total>

Reviewer #3: This is a very well written paper. I fully appreciate that authors fully document the limitations of both the study and the rapid test. These limitations in no way affect the importance to public health in the presentation of these data to the public.

6. PLOS authors have the option to publish the peer review history of their article (what does this mean?). If published, this will include your full peer review and any attached files.

Reviewer #1: No

Reviewer #2: No

Reviewer #3: No

---

## [Author Response · Author response to Decision Letter 0]

26 Oct 2021

Please see the Response to Reviewers document for additional information. Thank you for your consideration of our revised manuscript.

---

## [Decision Letter · Decision Letter 1]

18 Nov 2021

Evaluation of the Abbott BinaxNOW™ COVID-19 Test Ag Card for rapid detection of SARS-CoV-2 infection by a local public health district with a rural population

PONE-D-21-27586R1

Dear Dr. Pollreis,

We’re pleased to inform you that your manuscript has been judged scientifically suitable for publication and will be formally accepted for publication once it meets all outstanding technical requirements.

Kind regards,

Shinya Tsuzuki, MD, MSc

Academic Editor

PLOS ONE

Additional Editor Comments (optional):

Reviewers' comments:

Reviewer's Responses to Questions

**Comments to the Author**

1. If the authors have adequately addressed your comments raised in a previous round of review and you feel that this manuscript is now acceptable for publication, you may indicate that here to bypass the “Comments to the Author” section, enter your conflict of interest statement in the “Confidential to Editor” section, and submit your "Accept" recommendation.

Reviewer #2: All comments have been addressed

2. Is the manuscript technically sound, and do the data support the conclusions?

Reviewer #2: Yes

3. Has the statistical analysis been performed appropriately and rigorously? 

Reviewer #2: I Don't Know

4. Have the authors made all data underlying the findings in their manuscript fully available?

Reviewer #2: Yes

5. Is the manuscript presented in an intelligible fashion and written in standard English?

Reviewer #2: Yes

6. Review Comments to the Author

Reviewer #2: In the revised version of the manuscript the data still is presented very clearly and discussed in a balanced way. All reviewer comments were answered and/or worked into the manuscript. However, table numbering needs to be checked (tables 4 and 5 are now tables 3 and 4 ?)

7. PLOS authors have the option to publish the peer review history of their article (what does this mean?). If published, this will include your full peer review and any attached files.

Reviewer #2: No

---

## [Editor Report · Acceptance letter]

22 Nov 2021

PONE-D-21-27586R1 

Evaluation of the Abbott BinaxNOW COVID-19 Test Ag Card for rapid detection of SARS-CoV-2 infection by a local public health district with a rural population 

Dear Dr. Pollreis:

I'm pleased to inform you that your manuscript has been deemed suitable for publication in PLOS ONE. Congratulations! Your manuscript is now with our production department. 

Kind regards, 

on behalf of

Dr. Shinya Tsuzuki 

Academic Editor

PLOS ONE